# Neural nano-optics for high-quality thin lens imaging

Ethan Tseng [1,4], Shane Colburn[2,4], James Whitehead[2], Luocheng Huang[2], Seung-Hwan Baek[1], Arka Majumdar [2,3] & Felix Heide [1✉]

Nano-optic imagers that modulate light at sub-wavelength scales could enable new applications in diverse domains ranging from robotics to medicine. Although metasurface optics offer a path to such ultra-small imagers, existing methods have achieved image quality far worse than bulky refractive alternatives, fundamentally limited by aberrations at large apertures and low f-numbers. In this work, we close this performance gap by introducing a neural nano-optics imager. We devise a fully differentiable learning framework that learns a metasurface physical structure in conjunction with a neural feature-based image reconstruction algorithm. Experimentally validating the proposed method, we achieve an order of magnitude lower reconstruction error than existing approaches. As such, we present a high-quality, nano-optic imager that combines the widest field-of-view for full-color metasurface operation while simultaneously achieving the largest demonstrated aperture of 0.5 mm at an f-number of 2.

[1] Princeton University, Department of Computer Science, Princeton, NJ, USA. [2] University of Washington, Department of Electrical and Computer Engineering, Washington, WA, USA. [3] University of Washington, Department of Physics, Washington, WA, USA. [4]These authors contributed equally: Ethan Tseng and Shane Colburn. ✉email: fheide@princeton.edu

The miniaturization of intensity sensors in recent decades has made today's cameras ubiquitous across many application domains, including medical imaging, commodity smartphones, security, robotics, and autonomous driving. However, imagers that are an order of magnitude smaller could enable numerous novel applications in nano-robotics, in vivo imaging, AR/VR, and health monitoring. While sensors with submicron pixels do exist, further miniaturization has been prohibited by the fundamental limitations of conventional optics. Traditional imaging systems consist of a cascade of refractive elements that correct for aberrations, and these bulky lenses impose a lower limit on camera footprint. A further fundamental barrier is a difficulty of reducing focal length, as this induces greater chromatic aberrations.

We turn towards computationally designed metasurface optics (meta-optics) to close this gap and enable ultra-compact cameras that could facilitate new capabilities in endoscopy, brain imaging, or in a distributed fashion as collaborative optical "dust" on scene surfaces. Ultrathin meta-optics utilize subwavelength nano-antennas to modulate incident light with greater design freedom and space-bandwidth product over conventional diffractive optical elements (DOEs)[1–4]. Furthermore, the rich modal characteristics of meta-optical scatterers can support multifunctional capabilities beyond what traditional DOEs can do (e.g., polarization, frequency, and angle multiplexing). Meta-optics can be fabricated using widely available integrated circuits foundry techniques, such as deep ultraviolet lithography (DUV), without multiple etch steps, diamond turning, or grayscale lithography as used in polymer-based DOEs or binary optics.

Because of these advantages, researchers have harnessed the potential of meta-optics for building flat optics for imaging[5–7], polarization control[8], and holography[9]. Existing metasurface imaging methods, however, suffer from an order of magnitude higher reconstruction error than achievable with refractive compound lenses due to severe, wavelength-dependent aberrations that arise from discontinuities in their imparted phase[2,5,10–16]. Dispersion-engineering aims to mitigate this by exploiting group delay and group delay dispersion to focus broadband light[15–21], however, this technique is fundamentally limited to aperture designs of ~10s of microns[22]. As such, existing approaches have not been able to increase the achievable aperture sizes without significantly reducing the numerical aperture or supported wavelength range. Other attempted solutions only suffice for discrete wavelengths or narrowband illumination[11–14,23].

Metasurfaces also exhibit strong geometric aberrations that have limited their utility for wide field-of-view (FOV) imaging. Approaches that support wide FOV typically rely on either small input apertures that limit light collection[24] or use multiple metasurfaces[11], which drastically increases fabrication complexity. Moreover, these multiple metasurfaces are separated by a gap that scales linearly with the aperture, thus obviating the size benefit of meta-optics as the aperture increases.

Recently, researchers have leveraged computational imaging to offload aberration correction to post-processing software[10,25,26]. Although these approaches enable full-color imaging metasurfaces without stringent aperture limitations, they are limited to a FOV below 20° and the reconstructed spatial resolution is an order of magnitude below that of conventional refractive optics. Furthermore, existing learned deconvolution methods[27] have been restricted to variants of standard encoder-decoder architectures, such as the U-Net[28], and often fail to generalize to experimental measurements or handle large aberrations, as found in broadband metasurface imagers.

Researchers have proposed camera designs that utilize a single-optic instead of compound stacks[29,30], but these systems fail to match the performance of commodity imagers due to low diffraction efficiency. Moreover, the most successful approaches[29–31] hinder miniaturization because of their long back focal distances of more than 10 mm. Lensless cameras[32] instead reduce the size by replacing the optics with amplitude masks, but this severely limits spatial resolution and requires long acquisition times.

Recently, a variety of inverse design techniques have been proposed for meta-optics. Existing end-to-end optimization frameworks for meta-optics[7,33,34] are unable to scale to large aperture sizes due to prohibitive memory requirements and do not optimize for the final full-color image quality, often relying instead on intermediary metrics such as focal spot intensity.

In this work, we propose neural nano-optics, a high-quality, polarization-insensitive nano-optic imager for full-color (400 to 700 nm), wide FOV (40°) imaging with an f-number of 2. In contrast to previous works that rely on hand-crafted designs and reconstruction, we jointly optimize the metasurface and deconvolution algorithm with an end-to-end differentiable image formation model. The differentiability allows us to employ first-order solvers, which have been popularized by deep learning, for joint optimization of all parameters of the pipeline, from the design of the meta-optic to the reconstruction algorithm. The image formation model exploits a memory-efficient differentiable nano-scatterer simulator, as well as a neural feature-based reconstruction architecture. We outperform existing methods by an order of magnitude in reconstruction error outside the nominal wavelength range on experimental captures.

## Results

**Differentiable metasurface proxy model.** The proposed differentiable metasurface image formation model (Fig. 1e) consists of three sequential stages that utilize differentiable tensor operations: metasurface phase determination, PSF simulation and convolution, and sensor noise. In our model, polynomial coefficients that determine the metasurface phase are optimizable variables, whereas experimentally calibrated parameters characterizing the sensor readout and the sensor-metasurface distance are fixed.

The optimizable metasurface phase function $\phi$ as a function of distance $r$ from the optical axis is given by

$$\phi(r) = \sum_{i=0}^{n} a_i \left(\frac{r}{R}\right)^{2i},  \tag{1}$$

where $\{a_0, \ldots a_n\}$ are optimizable coefficients, $R$ is the phase mask radius, and $n$ is the number of polynomial terms. We optimize the metasurface in this phase function basis as opposed to in a pixel-by-pixel manner to avoid local minima. The number of terms $n$ is user-defined and can be increased to allow for finer control of the phase profile, in the experiments we used $n = 8$. We used even powers in the polynomial to impart a spatially symmetric PSF in order to reduce the computational burden, as this allows us to simulate the full FOV by only simulating along one axis. This phase, however, is only defined for a single, nominal design wavelength, which is a fixed hyperparameter set to 452 nm in our optimization. While this mask alone is sufficient for modeling monochromatic light propagation, we require the phase at all target wavelengths to design for a broadband imaging scenario.

To this end, at each scatterer position in our metasurface, we apply two operations in sequence. The first operation is an inverse, phase-to-structure mapping that computes the scatterer geometry given the desired phase at the nominal design wavelength. With the scatterer geometry determined, we can then apply a forward, structure-to-phase mapping to calculate the phase at the remaining target wavelengths. Leveraging an effective index approximation that ensures a unique geometry for each phase shift in the 0 to $2\pi$ range, we ensure differentiability, and

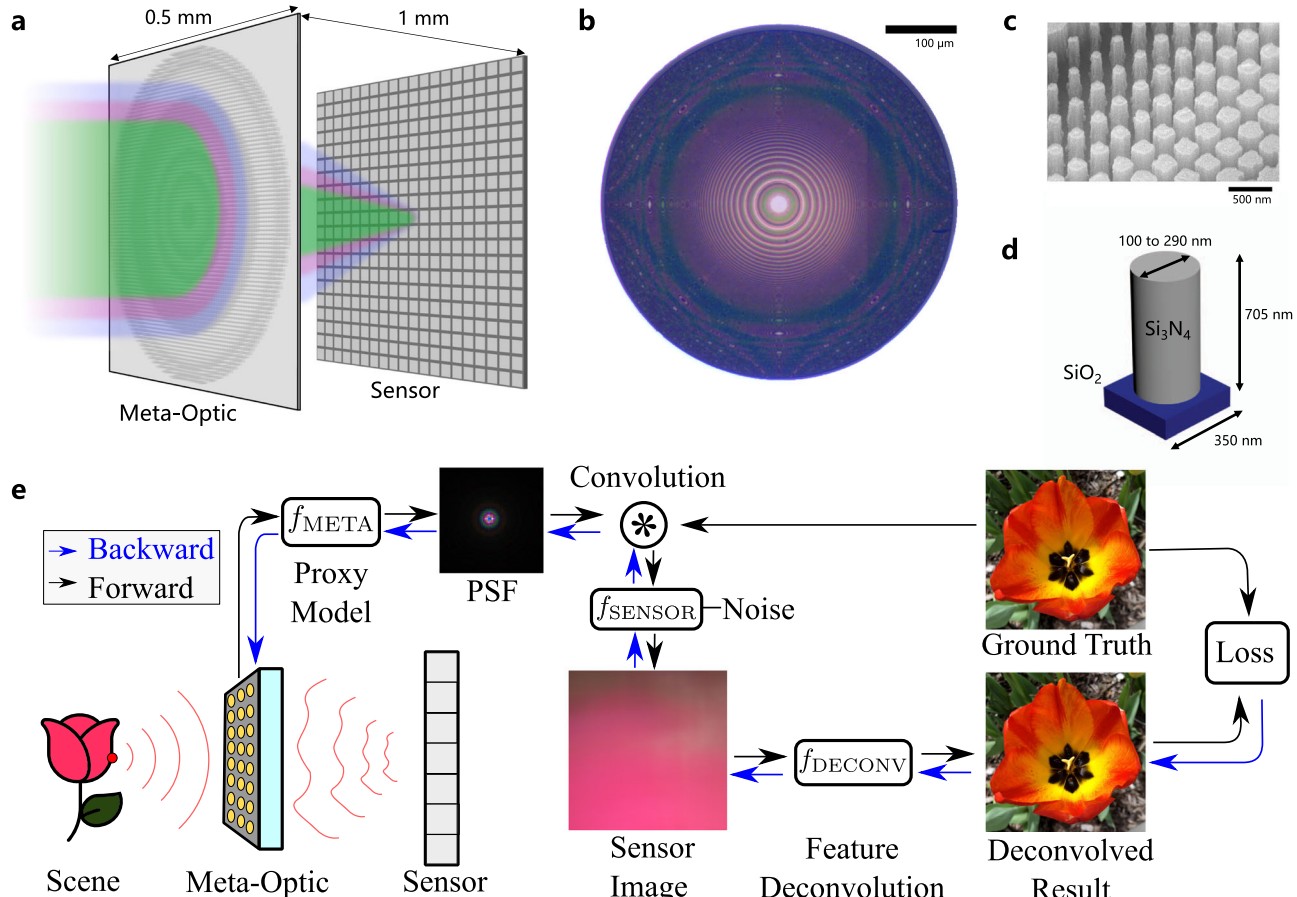

**Fig. 1 Neural nano-optics end-to-end design.** Our learned, ultrathin meta-optic as shown in (**a**) is 500 μm in thickness and diameter, allowing for the design of a miniature camera. The manufactured optic is shown in (**b**). A zoom-in is shown in (**c**) and nanopost dimensions are shown in (**d**). Our end-to-end imaging pipeline shown in **e** is composed of the proposed efficient metasurface image formation model and the feature-based deconvolution algorithm. From the optimizable phase profile, our differentiable model produces spatially varying PSFs, which are then patch-wise convolved with the input image to form the sensor measurement. The sensor reading is then deconvolved using our algorithm to produce the final image. The illustrations above "Meta-Optic" and "Sensor" in (**e**) were created by the authors using Adobe Illustrator.

can directly optimize the phase coefficients by adjusting the scatterer dimensions and computing the response at different target wavelengths. See Supplementary Note 4 for details.

These phase distributions differentiably determined from the nano-scatterers allow us to then calculate the PSF as a function of wavelength and field angle to efficiently model full-color image formation over the whole FOV, see Supplementary Fig. 3. Finally, we simulate sensing and readout with experimentally calibrated Gaussian and Poisson noise by using the reparameterization and score-gradient techniques to enable backpropagation, see Supplementary Note 4 for a code example.

While researchers have designed metasurfaces by treating them as phase masks[5,35], the key difference between our approach and previous ones is that we formulate a proxy function that mimics the phase response of a scatterer under the local phase approximation, enabling us to use automatic differentiation for inverse design.

When compared directly against alternative computational forward simulation methods, such as finite-difference time-domain (FDTD) simulation[33], our technique is approximate but is more than three orders of magnitudes faster and more memory-efficient. For the same aperture as our design, FDTD simulation would require the order of 30 terabytes for accurate meshing alone. Our technique instead only scales quadratically with length. This enables our entire end-to-end pipeline to achieve a memory reduction of over 3000×, with metasurface

simulation and image reconstruction both fitting within a few gigabytes of GPU RAM.

The simulated and experimental phase profiles are shown in Figs. 1b and 3. Note that the phase changes rapidly enough to induce aliasing effects in the phase function; however, since the full profile is directly modeled in our framework these effects are all incorporated into the simulation of the structure itself and are accounted for during optimization.

**Neural feature propagation and learned nano-optics design.** We propose a neural deconvolution method that incorporates learned priors while generalizing to unseen test data. Specifically, we design a neural network architecture that performs deconvolution on a learned feature space instead of on raw image intensity. This technique combines both the generalization of model-based deconvolution and the effective feature learning of neural networks, allowing us to tackle image deconvolution for meta-optics with severe aberrations and PSFs with a large spatial extent. This approach generalizes well to experimental captures even when trained only in simulation.

The proposed reconstruction network architecture comprises three stages: a multi-scale feature extractor $f_{FE}$, a propagation stage $f_{Z \rightarrow W}$ that deconvolves these features (i.e., propagates features Z to their deconvolved spatial positions W), and a decoder stage $f_{DE}$ that combines the propagated features into a

final image. Formally, our feature propagation network performs the following operations:

$$\text{O} = f_{\text{DE}}\big(\underset{\uparrow}{f_{\text{Z}\to\text{W}}}\big(\underset{\uparrow}{f_{\text{FE}}}(\text{I}),\ \text{PSF}\big)\big), \qquad (2)$$

$$\underset{\text{Decoder}}{} \qquad \underset{\text{Feature Extraction}}{}$$

where I is the raw sensor measurement and O is the output image.

Both the feature extractor and decoder are constructed as fully convolutional neural networks. The feature extractor identifies features at both the native resolution and multiple scales to facilitate learning low-level and high-level features, allowing us to encode and propagate higher-level information beyond raw intensity. The subsequent feature propagation stage $f_{\text{Z}\to\text{W}}$ is a deconvolution method that propagates the features Z to their inverse-filtered positions W via a differentiable mapping such that W is differentiable with respect to Z. Finally, the decoder stage then converts the propagated features back into image space, see Supplementary Note 5 for architecture details. When compared against existing state-of-the-art deconvolution approaches we achieve over 4 dB Peak signal-to-noise ratio (PSNR) improvement (more than 2.5× reduction in mean squared error) for deconvolving challenging metasurface incurred aberrations, see Supplementary Table 11.

Both our metasurface image formation model and our deconvolution algorithm are incorporated into a fully differentiable, end-to-end imaging chain. Our metasurface imaging pipeline allows us to apply first-order stochastic optimization methods to learn metasurface phase parameters $\mathcal{P}_{\text{META}}$ and parameters $\mathcal{P}_{\text{DECONV}}$ for our deconvolution network $f_{\text{DECONV}}$ that will minimize our endpoint loss function $\mathcal{L}$, which in our case is a perceptual quality metric. Our image formation model is thus defined as

$$\mathbf{O} = f_{\text{DECONV}}\big(\mathcal{P}_{\text{DECONV}}, f_{\text{SENSOR}}\big(\mathbf{I} * f_{\text{META}}(\mathcal{P}_{\text{META}})\big), f_{\text{META}}(\mathcal{P}_{\text{META}})\big) \qquad (3)$$

where I is an RGB training image, $f_{\text{META}}$ generates the metasurface PSF from $\mathcal{P}_{\text{META}}$, $*$ is convolution, and $f_{\text{SENSOR}}$ models the sensing process including sensor noise. Since our deconvolution method is non-blind, $f_{\text{DECONV}}$ takes in $f_{\text{META}}(\mathcal{P}_{\text{META}})$. We then solve the following optimization problem

$$\{\mathcal{P}^*_{\text{META}}, \mathcal{P}^*_{\text{DECONV}}\} = \underset{\mathcal{P}_{\text{META}}, \mathcal{P}_{\text{DECONV}}}{\arg\min} \sum_{i=1}^{M} \mathcal{L}(\mathbf{O}^{(i)}, \mathbf{I}^{(i)}). \qquad (4)$$

The final learned parameters $\mathcal{P}^*_{\text{META}}$ are used to manufacture the meta-optic and $\mathcal{P}^*_{\text{DECONV}}$ determines the deconvolution algorithm, see Supplementary Note 4 for further details.

**Imaging demonstration**. High-quality, full-color image reconstructions using our neural nano-optic are shown in Fig. 2 and in Supplementary Figs. 19, 20, 21, 22, 23. We perform comparisons against a traditional hyperbolic meta-optic designed for 511 nm and the state-of-the-art cubic meta-optic from Colburn et al.[10]. Additional experimental comparisons against alternative single-optic and meta-optic designs are shown in Supplementary Note 11. Ground truth images are acquired using a six-element compound optic that is 550,000× larger in volume than the meta-optics. Our full computational reconstruction pipeline runs at real-time rates and requires only 58 ms to process a 720 px × 720 px RGB capture.

The traditional hyperbolic meta-optic experiences severe chromatic aberrations at larger and shorter wavelengths. This is observed in the heavy red blurring in Fig. 2a and the washed-out

blue color in Fig. 2c. The cubic meta-optic maintains better consistency across color channels but suffers from artifacts owing to its large, asymmetric PSF. In contrast, we demonstrate high-quality images without these aberrations, which are observable in the fine details in the fruits in Fig. 2a, the patterns on the lizard in Fig. 2b, and the flower petals in Fig. 2c. We quantitatively validate the proposed neural nano-optic by measuring reconstruction error on an unseen test set of natural images, on which we obtain 10× lower mean squared error than existing approaches, see Supplementary Table 12. In addition to natural image reconstruction, we also measured the spatial resolution using standard test charts, see Supplementary Note 10. Our nano-optic imager achieves a spatial resolution of 214 lp/mm across all color channels at 120 mm object distance. We improve spatial resolution by an order of magnitude over the previous state-of-the-art by Colburn et al.[10] which achieved 30 lp/mm.

**Characterizing nano-optics performance**. Through our optimization process, our meta-optic learns to produce compact PSFs that minimize chromatic aberrations across the entire FOV and across all color channels. Unlike designs that exhibit a sharp focus for a single wavelength but significant aberrations at other wavelengths, our optimized design strikes a balance across wavelengths to facilitate full-color imaging. Furthermore, the learned meta-optic avoids the large PSFs used previously by Colburn et al.[10] for computational imaging.

After optimization, we fabricated our neural nano-optics (Fig. 3), as well as several heuristic designs for a comparison. Note that commercial large-scale production of meta-optics can be performed using high-throughput processes based on DUV lithography which is standard for mature industries such as semiconductor integrated circuits, see Supplementary Note 3 for details. The simulated and experimental PSFs are shown in Fig. 3 and are in strong agreement, validating the physical accuracy of the proxy metasurface model. To account for manufacturing imperfections, we perform a PSF calibration step where we capture the PSFs using the fabricated meta-optics. We then finetune our deconvolution network by replacing the proxy-based metasurface simulator with the captured PSFs. The finetuned network is deployed on experimental captures using the setup shown in Supplementary Fig. 7. This finetuning calibration step does not train on experimental captures, we only require the measured PSFs. Thus, we do not require the experimental collection of a vast image dataset.

We observe that the PSF for our optimized meta-optic exhibits a combination of the compact shape and minimal variance across field angles, as expected for our design. PSFs for a traditional hyperbolic meta-optic (511 nm) instead have significant spatial variation across field angles and severe chromatic aberrations that cannot be compensated through deconvolution. While the cubic design from Colburn et al.[10] does exhibit spatial invariance, its asymmetry and large spatial extent introduce severe artifacts that reduce image quality. See Fig. 3 and Supplementary Note 8 for comparisons of the traditional meta-optic and Colburn et al.[10] against ours. We also show corresponding modulation transfer functions (MTFs) for our design in Fig. 3. The MTF does not change appreciably with incidence angle and also preserves a broad range of spatial frequencies across the visible spectrum.

**Discussion**

In this work, we present an approach for achieving high-quality, full-color, wide FOV imaging using neural nano-optics. Specifically, the proposed learned imaging method allows for an order of magnitude lower reconstruction error on experimental data than existing works. The key enablers of this result are our

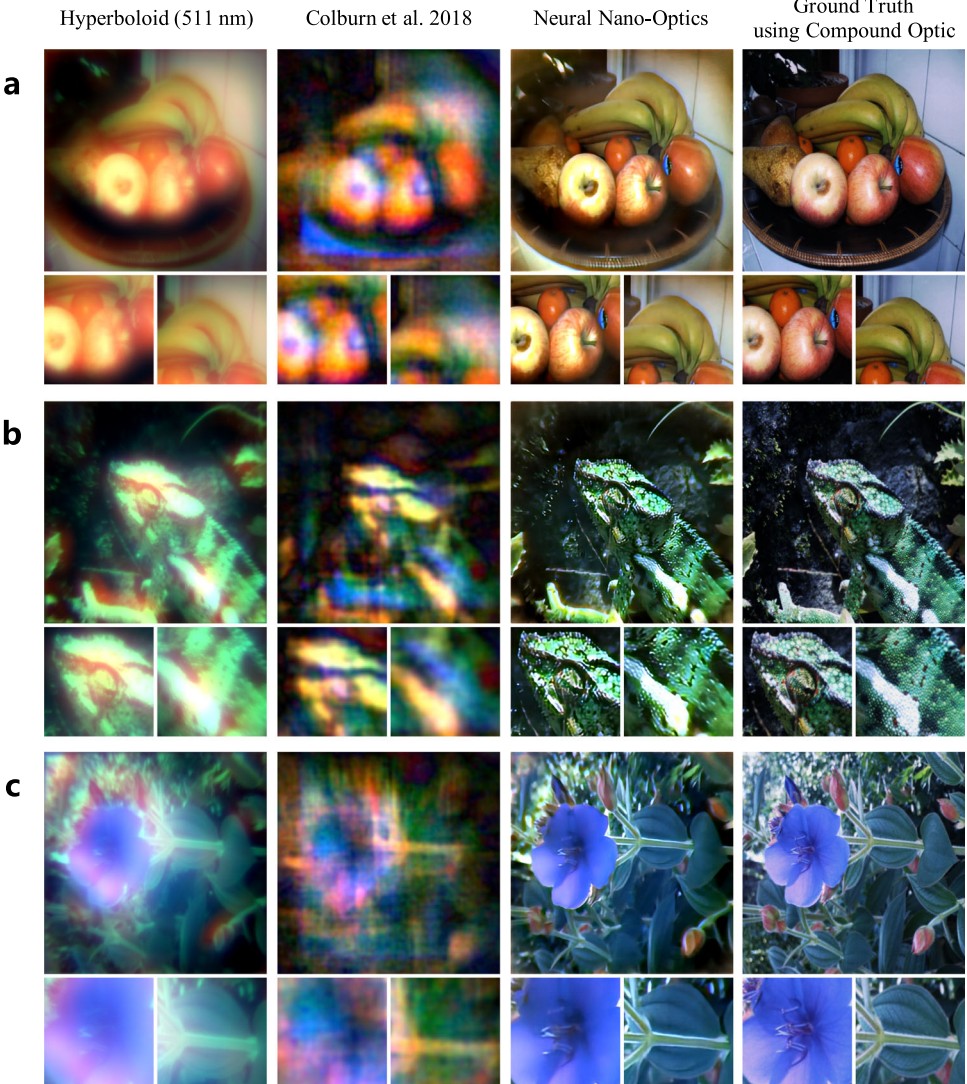

**Fig. 2 Experimental imaging results.** Compared to existing state-of-the-art designs, the proposed neural nano-optic produces high-quality wide FOV reconstructions corrected for aberrations. Example reconstructions are shown for a still life with fruits in (**a**), a green lizard in (**b**), and a blue flower in (**c**). Insets are shown below each row. We compare our reconstructions to ground truth acquisitions using a high-quality, six-element compound refractive optic, and we demonstrate accurate reconstructions even though the volume of our meta-optic is 550,000× lower than that of the compound optic.

differentiable meta-optical image formation model and novel deconvolution algorithm. Combined together as a differentiable end-to-end model, we jointly optimize the full computational imaging pipeline with the only target metric being the quality of the deconvolved RGB image—sharply deviating from existing methods that penalize focal spot size in isolation from the reconstruction method.

We have demonstrated the viability of meta-optics for high-quality imaging in full-color, over a wide FOV. No existing meta-optic demonstrated to date approaches a comparable combination of image quality, large aperture size, low $f$-number, wide fractional bandwidth, wide FOV, and polarization insensitivity (see Supplementary Notes 1 and 2), and the proposed method could scale to mass production. Furthermore, we demonstrate image quality on par with a bulky, six-element commercial compound lens even though our design volume is 550,000× lower and utilizes a single metasurface.

We have designed neural nano-optics for a dedicated, aberration-free imaging task, but we envision extending our work towards flexible imaging with reconfigurable nanophotonics for diverse tasks, ranging from an extended depth of field to classification or object detection tasks. We believe that the proposed method takes an essential step towards ultra-small cameras that may enable novel applications in endoscopy, brain imaging, or in a distributed fashion on object surfaces.

## Methods

**Optimization**. We used TensorFlow 2 to design and evaluate our neural nano-optic. See Supplementary Note 6 for details on the training procedure, hyper-parameters, and loss functions. We used the INRIA Holiday dataset for training[36].

**Sample fabrication**. Beginning with a double side polished fused silica wafer, we deposit 705 nm of silicon nitride via plasma-enhanced chemical vapor deposition to form our device layer. We then spin coat with ZEP 520A resist and sputter an 8 nm gold charge dissipation layer followed by exposure with a JEOL JBX6300FS electron-beam lithography system at 100 kV and 8 nA. After stripping the gold, we develop amyl acetate followed by immersion in isopropyl alcohol. To define the etch mask, we evaporate 50 nm of aluminum and lift off via sonication in methylene chloride, acetone, and isopropyl alcohol. We then etch the silicon nitride layer using $CHF_3$ and $SF_6$ chemistry with an inductively coupled plasma etcher. Following stripping of the aluminum etch mask, we coat and pattern AZ 1512 photoresist on the chip, followed by aluminum evaporation and lift-off in order to define a hard aperture to block stray light.

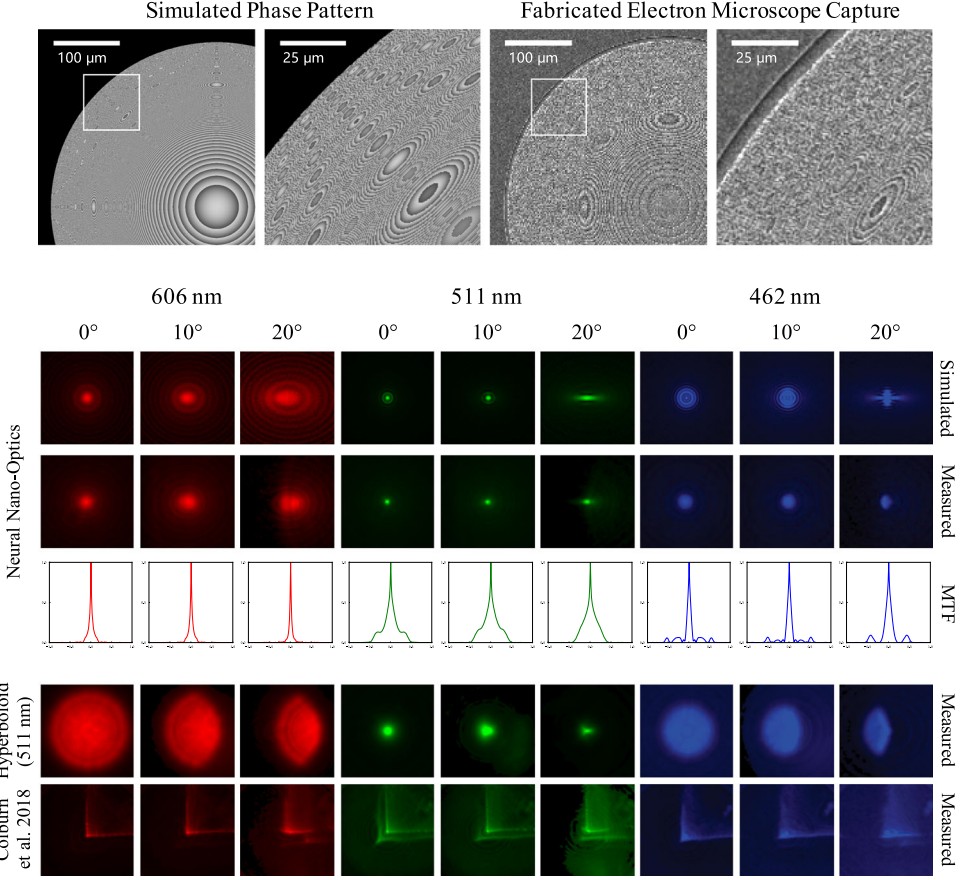

**Fig. 3 Meta-optics characterization.** The proposed learned meta-optic is fabricated using electron-beam lithography and dry etching, and the corresponding measured PSFs, simulated PSFs, and simulated MTFs are shown. Before capturing images, we first measure the PSFs of the fabricated meta-optics to account for deviations from the simulation. Nevertheless, the match between the simulated PSFs and the measured PSFs validates the accuracy of our metasurface proxy model. The proposed learned design maintains consistent PSF shape across the visible spectrum and for all field angles across the FOV, facilitating downstream image reconstruction. In contrast, the PSFs of the traditional meta-optic and the cubic design proposed by Colburn et al.[10] both exhibit severe chromatic aberrations. The red (606 nm) and blue (462 nm) PSFs of the traditional meta-optic are defocused and change significantly across the FOV. The PSFs for the cubic design exhibit long tails that leave post-deconvolution artifacts.

**Experimental setup**. After fabrication of the meta-optic, we account for fabrication error by performing a PSF calibration step. This is accomplished by using an optical relay system to image a pinhole illuminated by fiber-coupled LEDs. We then conduct imaging experiments by replacing the pinhole with an OLED monitor. The OLED monitor is used to display images that will be captured by our nano-optic imager. See Supplementary Note 7 for details.

## Data availability

The raw capture data is available at https://doi.org/10.5281/zenodo.5637678.

## Code availability

The code used to design and evaluate the neural nano-optic is available at https://doi.org/10.5281/zenodo.5637678.

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

## Acknowledgements

This research was supported by NSF-1825308, DARPA (Contract no. 140D0420C0060), the UW Reality Lab, Facebook, Google, Futurewei, and Amazon. A.M. is also supported by a Washington Research Foundation distinguished investigator award. Part of this work was conducted at the Washington Nanofabrication Facility/Molecular Analysis Facility, a National Nanotechnology Coordinated Infrastructure (NNCI) site at the University of Washington with partial support from the National Science Foundation via awards NNCI-2025489 and NNCI-1542101.

## Author contributions

E.T. and F.H. developed the feature space deconvolution technique, integrated the metasurface model and deconvolution framework, performed the final design optimizations, and led the manuscript writing. S.C. developed the differentiable metasurface and sensor noise model, led the experiment, and assisted E.T. in writing the manuscript. J.W. fabricated all the devices and assisted in the experiment. L.H. developed the scripts for automated image capture and assisted in the experiment. S.-H.B. assisted in writing the manuscript. A.M. and F.H. supervised the project and assisted in writing the manuscript.

## Competing interests

A.M. is cofounder of Tunoptix Inc., which is commercializing technology discussed in this manuscript. S.C. conducted the work in this manuscript while at the University of Washington and is now at Tunoptix Inc.
