## [Peer Review File · Nature Communications]

Neural Nano-Optics for High-quality Thin Lens ImagingREVIEWER COMMENTS

Reviewer #1 (Remarks to the Author):

In the submitted manuscript, the authors have described the joint design of a meta-optic along with the parameters of a convolutional neural network image formation method to be used with the meta-optic. With the disclaimer that I am not knowledgeable enough to know whether the comparisons in Figure 2 are the best possible comparisons, I find the results extraordinarily impressive, and I am sold on the concept that the authors have made a substantial contribution toward the promise of metasurfaces for extreme miniaturization of full optical imagers. I recommend publication in Nature Communications after suitable improvements of the manuscript and supplementary document.

The manuscript is very well illustrated, but the writing has some room for improvement. Though the only "errors" that I will point out are trivial, I hope that the authors will also consider the weaknesses that I describe. I think that some improvements will broaden the appreciation for and impact of this work.

"differentiable": Counting the captions and author contributions, the word "differentiable" occurs 12 times in the article file. Yet, in my opinion, there is no effort to help the reader understand the significance of differentiability. Yes, many readers will understand without being told, but I think this is a major weakness nevertheless. As a lesser criticism, please carefully consider what this adjective is modifying. I think that it is clear when applied to "model" but much less clear when applied to "method," and the first use (in the abstract) makes for a difficult to understand sentence.

Metasurface proxy model: Though I understand your achievement to be broad (conceptualization, design, fabrication, testing, etc.), it seems that the greatest novelty is in having a representation for the phase response of the metasurface that -- because of its simple form -- can be part of an end-to-end optimization that is joint with the reconstruction algorithm. Because of this centrality, I recommend that the section starting on line 80 be expanded a bit. How does Eq. (1) relate to what is seen in Figure 1(b) and the top row of Figure 3? Those figures do not seem to show a dependence only on r . Why are there only even powers in the polynomial? Since r is positive, having even powers is not for positivity. Why do you not specify the number of terms in the sum, leaving that only for supplemental material. Furthermore, it seems that there was not enough care in making the manuscript (lines 86-93) consistent with the supplemental document (lines 102-112), or for the relationships to be clear.

I think the manuscript would benefit from a summary of the parameters that are optimized (separating hardware and reconstruction algorithm), including the numbers of these parameters.

Stylistic suggestions and minor corrections:

line 19: Improve the parsability of this sentence. It currently reads as the largest aperture among those that are 0.5 mm, f/2.

line 35: I think that "achieve" for a negative characteristic is a bit strange.

line 38: Ungrammatical.

line 51: "a FOV" should be "an FOV".

line 79: Finally what?

line 94: I think that you apply two operations in sequence rather than two sequential operations.

line 135: PSNR is undefined.

line 156: reconstruciton

-Vivek Goyal

Reviewer #2 (Remarks to the Author):

The paper presents a new optimization technique for meta-material based lens fabrication. This is a very active area of research and there is significant related work on multiple different approaches for

creating meta-material thin lenses. The key innovation in the proposed approach is the neural network based design of the computational algorithm for image reconstruction.

The idea is interesting and has some merits. But there are many important details and comparisons that are missing that eventually detract from the impact of this paper.

a. Is there an actual application domain, where the proposed metamaterial based design results in the best possible performance for that application? The paper is interesting technically --- but doesn't compare with non meta-material based potential alternatives at all. Given that there are no application dependent results shown, my suspicion is that in almost all of these application domains existing other techniques such as GRIN lenses, waveguides, micro-optics, conventional fresnel lenses or similar may actually be better. In essence, the point this paper produces a better tradeoff in metamaterial lens design is a technical advance but it would be important to understand whether that is consequential in terms of any application domain as it stands today.

b. The central idea of the manuscript is the optimization of metasurface phase function for broadband, large FOV operation. The manuscript claims that the presented work is an order of magnitude better than existing meta-optics. However, the key question that remains to be addressed is whether metasurface-based imaging is necessary for the proposed application. The same optimization on a diffractive lens (like Fresnel's lens) would perhaps outperform metasurface-based imaging. Diffractive lens made of materials like glass does not suffer from severe chromatic aberrations like metasurfaces. The authors should compare their design with an optimization carried out on a diffractive lens.

c. Meta-elements impart wavelength-dependent phase. Equation 1 does not specify the wavelength at which the phase function is defined.

d. The manuscript states that a novelty of the presented work is bypassing the need for complex FDTD simulations. However, the manuscript describes only a technique to find the optimum phase distribution on a metasurface. The phase distribution cannot be mapped exactly to a structure without FDTD simulations. Such mapping of geometry to a phase was performed even in the earliest metasurface papers. And, the current work also takes the same or similar approach.

e. Ablation study -- the paper doesn't present a detailed ablation study that allows one to better understand the impact of the 'design stage' from the image enhancement that was obtained through neural networks. In particular, the comparisons shown are somewhat misleading, the prior works don't benefit from any advanced image enhancement technologies such as neural networks while the images from

this paper do...its important to understand whether the vast majority of the improvements are coming from the design -- i.e., the new meta-material lens, or the use of neural networks for image reconstruction. One can only understand this if a proper well designed ablation study is presented.

Reviewer #3 (Remarks to the Author):

In this work, the authors propose a differentiable learning method that learns a metasurface's physical structure in conjunction with a neural feature-based image reconstruction algorithm. They also experimentally verify the proposed method, and present the high-quality, nano-optic imager that combines the wide field of view for full-color metasurface operation with high numerical aperture. The proposed idea takes the advantages of both metasurface devices and the neural feature based reconstruction method, the experimental results agree well with their calculations. I would like to support its publication in NC after revisions.

Some comments:

1. It might be not accurate to say "the first neural nano-optics", nano-optics is a subject, is a research area, not a device.
2. There are quite a lot of descriptions which may be wrong, for example, "first", "widest". The numerical aperture (the author use f-number) of the metalens could be very high.
3. The wavelength dependent focal efficiency should be measured in the visible spectral regime.

Point-by-point Response: “Neural Nano-Optics for High-quality Thin Lens Imaging.”

Comment: With the disclaimer that I am not knowledgeable enough to know whether the comparisons in Figure 2 are the best possible comparisons, I find the results extraordinarily impressive, and I am sold on the concept that the authors have made a substantial contribution toward the promise of metasurfaces for extreme miniaturization of full optical imagers. I recommend publication in Nature Communications after suitable improvements of the manuscript and supplementary document. The manuscript is very well illustrated, but the writing has some room for improvement. Though the only "errors" that I will point out are trivial, I hope that the authors will also consider the weaknesses that I describe. I think that some improvements will broaden the appreciation for and impact of this work.

Response: We thank the reviewer for their detailed feedback and are pleased they found that our work represents a substantial contribution to the field. We have now incorporated the suggested improvements in the revised manuscript.

Comment: "differentiable": Counting the captions and author contributions, the word "differentiable" occurs 12 times in the article file. Yet, in my opinion, there is no effort to help the reader understand the significance of differentiability. Yes, many readers will understand without being told, but I think this is a major weakness nevertheless. As a lesser criticism, please carefully consider what this adjective is modifying. I think that it is clear when applied to "model" but much less clear when applied to "method," and the first use (in the abstract) makes for a difficult to understand sentence.

Response: We agree that “differentiable”, in the machine-learning sense, should be used more carefully and we have incorporated this suggestion in our revised manuscript. Specifically, at the first mention of “differentiable” in the introduction we now provide additional exposition that explains why “differentiability” is important and how it is used to jointly optimize the entire meta-optic imaging pipeline using first order solvers. Secondly, we also now provide additional explanation of “differentiable” each time it is used as a modifier for “method”, such as when the feature propagator is introduced.

Comment: Metasurface proxy model: Though I understand your achievement to be broad (conceptualization, design, fabrication, testing, etc.), it seems that the greatest novelty is in having a representation for the phase response of the metasurface that -- because of its simple form -- can be part of an end-to-end optimization that is joint with the reconstruction algorithm. Because of this centrality, I recommend that the section starting on line 80 be expanded a bit. How does Eq. (1) relate to what is seen in Figure 1(b) and the top row of Figure 3? Those figures do not seem to show a dependence only on r .

Response: We have incorporated additional explanation in that section to further illustrate the relationship between Equation 1 and the observed phase profile of the metasurface optic. The reviewer raises a good point that the meta-optic shown in Figure 1 appears to have a structure that is not completely radially symmetric. This, however, arises from a sampling of the phase profile onto the lattice formed by the subwavelength-spaced scatterers. In this case, the phase changes rapidly enough to induce aliasing effects in the phase function; however, since the full profile is directly modelled in our framework, these effects are all incorporated into the simulation of the structure itself and are accounted for during optimization. Since traversing a path along one of the lattice’s axes versus along a diagonal entails passing through a different number of discrete sample points, under this

condition the meta-optic can appear to not be rotationally symmetric even when parameterized with radial polynomials.

Comment: Why are there only even powers in the polynomial? Since r is positive, having even powers is not for positivity.

Response: We used radial powers in order to impart a radially symmetric PSF which reduced the computational burden, as this allows us to simulate the full FOV by simulating only one axis. While we could have included odd terms as well or other polynomial parameterizations, the selection of even powers was to be consistent with the form of an even asphere which is commonly used in lens designs. We have now added revisions to the manuscript that clarify this point.

Comment: Why do you not specify the number of terms in the sum, leaving that only for supplemental material.

Response: The number of terms is user defined and can be increased to allow for finer control of the phase profile. Nevertheless, we found that using 8 terms provided sufficient control over the phase to allow for high-quality design. We have now added revisions that clarify this point.

Comment: Furthermore, it seems that there was not enough care in making the manuscript (lines 86-93) consistent with the supplemental document (lines 102-112), or for the relationships to be clear.

Response: We have further revised the manuscript to connect these two sections. Specifically, we use $\phi(r)$ in both Eq. 1 of the main manuscript and Eq. S1 of the Supplemental Document to indicate the same quantity, the phase at distance r away from the optical axis for the nominal wavelength (452 nm). This is all now described in the explanation text for Eq. S1. We have also rewritten equations S1 and S2 for consistency.

Comment: I think the manuscript would benefit from a summary of the parameters that are optimized (separating hardware and reconstruction algorithm), including the numbers of these parameters.

Response: We agree that this would add greater clarity to the manuscript. We described this in Figure S5 where different parameters are highlighted with different colors, and we have incorporated an additional summary of the parameters in the Supplemental Document at the end of Section "Differentiable Proxy-Based Metasurface Image Formation" and in Table S5.

Comment: Stylistic suggestions and minor corrections in the manuscript.

Response: We thank the reviewer for the detailed suggestions. All editorial and minor corrections have been incorporated into the revised manuscript.

Comment: Is there an actual application domain, where the proposed metamaterial based design results in the best possible performance for that application? The paper is interesting technically --- but doesn't compare with non meta-material based potential alternatives at all. Given that there are no application dependent results shown, my suspicion is that in almost all of these application domains existing other techniques such as GRIN lenses, waveguides, micro-optics, conventional

fresnel lenses or similar may actually be better. In essence, the point this paper produces a better tradeoff in metamaterial lens design is a technical advance but it would be important to understand whether that is consequential in terms of any application domain as it stands today.

Response: The reviewer raises a good point regarding whether our proposed meta-optic design platform could produce results superior to alternatives in certain application domains. There are indeed alternative approaches based on GRIN lenses, Fresnel optics, and others; however, these all have their own associated challenges as well. While other optic platforms would require additional elements to correct for aberrations but because of the subwavelength pitch of the metasurface, it inherently has a higher space-bandwidth product and possesses far more degrees of freedom for manipulating wavefronts compared to the other conventional alternatives that the reviewer mentioned^{1,2}. In some cases, as shown in the manuscript, this enables us to collapse the functionality of multiple optics into a single profile that would typically require multiple elements. This, in combination with our reconstruction software, enables image quality on par with that of traditional refractive compound lenses as shown in the main text of our manuscript. Another example of the capability of meta-optics compared to alternatives is how traditional binary optics are incapable of imparting such high-resolution phase profiles, limiting their f-number. Furthermore, there is a strong case to be made for meta-optics reducing cost and complexity of optical systems even if they did not contribute to large enhancements in performance, given that reducing the number of surfaces can mitigate alignment complexity, vibration sensitivity, and manufacturing complexity.

Comment: The central idea of the manuscript is the optimization of metasurface phase function for broadband, large FOV operation. The manuscript claims that the presented work is an order of magnitude better than existing meta-optics. However, the key question that remains to be addressed is whether metasurface-based imaging is necessary for the proposed application. The same optimization on a diffractive lens (like Fresnel's lens) would perhaps outperform metasurface-based imaging. Diffractive lens made of materials like glass does not suffer from severe chromatic aberrations like metasurfaces. The authors should compare their design with an optimization carried out on a diffractive lens.

Response: We now include additional arguments into the manuscript that describe the advantages of metasurfaces over diffractive optics, including but not limited to broadband, large FOV color imaging. Specifically: (1) Higher numerical apertures can be achieved as metasurfaces can sample a higher resolution phase profile. (2) Drastically simplified fabrication. Metasurfaces primarily require only one lithography stage whereas DOEs often require multi-level etching, i.e. four stages with different etch depth for 16 levels. (3) Metasurfaces offer richer characteristics for manipulating spectrum and polarization. While these applications are not the focus of this manuscript, demonstrating metasurface imaging for full color imaging could enable these applications in the future.

Comment: Meta-elements impart wavelength-dependent phase. Equation 1 does not specify the wavelength at which the phase function is defined.

Response: We have added additional clarification that the wavelength for Equation 1 is for a nominal wavelength of 452 nm. These coefficients define the phase at that wavelength but the associated proxy model enables us to then compute the phase at other wavelengths used in the optimization given the coefficients defining the phase at the nominal wavelength.

Comment: The manuscript states that a novelty of the presented work is bypassing the need for complex FDTD simulations. However, the manuscript describes only a technique to find the optimum phase distribution on a metasurface. The phase distribution cannot be mapped exactly to a structure without FDTD simulations. Such mapping of geometry to a phase was performed even in the earliest metasurface papers. And, the current work also takes the same or similar approach.

Response: The reviewer is correct that to capture the totality of the underlying physics in a meta-optical scatterer, we would need to use a method such as FDTD simulation. FDTD simulation, however, is very costly and it is standard in the metasurface community to abstract the response of scatterer using the local phase approximation by modelling its response using a pre-computed transmission coefficient (e.g., calculated via rigorous coupled-wave analysis or via FDTD). As such, using FDTD iteratively in our method for thousands of iterations is infeasible. The reviewer is also correct that previous papers have designed metasurfaces by treating them as phase masks. The key difference between our approach and previous ones is that we formulate a proxy function that mimics the phase response of a scatterer under the local phase approximation, enabling us to use automatic differentiation for inverse design. While many metasurface designs make a local phase approximation in which the scatterer's response is abstracted in terms of phase coefficient, to the best of our knowledge, our work is the first to explicitly fit this response and include it within an automatic differentiation framework.

Comment: Ablation study -- the paper doesn't present a detailed ablation study that allows one to better understand the impact of the 'design stage' from the image enhancement that's obtained through neural networks. In particular, the comparisons shown are somewhat misleading, the prior works don't benefit from any advanced image enhancement technologies such as neural networks while the images from this paper do...it's important to understand whether the vast majority of the improvements are coming from the design -- i.e., the new meta-material lens, or the use of neural networks for image reconstruction. One can only understand this if a proper well designed ablation study is presented.

Response: We present these ablation comparisons in the Supplemental Document. Specifically, Section "Validation of Neural Nano-Optic Design" ablates the meta-optic design while keeping the deconvolution method fixed. We apply the proposed neural feature propagation deconvolution to the cubic, log-asphere, hyperboloid designs and we demonstrate that the optimized design performs the best, both in simulation and in experimental demonstration. Please see Table S10 and Figures S9 and S10. We also provide other comparisons between different meta-optic designs such as the PSFs (Figure S12), MTFs (Figure S13), cross-sectional intensity profiles (Figure S14), and efficiency (Figure S15).

In Section "Validation of Neural Feature Propagation", we ablate the deconvolution method. We apply different deconvolution algorithms to the cubic, log-asphere, and optimized designs and we demonstrate that the neural feature propagation deconvolution achieves the highest reconstruction performance regardless of the metasurface optical design. Please see Table S11 and Figures S16, S17, S18.

Comment: It might be not accurate to say "the first neural nano-optics", nano-optics is a subject, is a research area, not a device.

Response: We thank the reviewer for this comment. We have revised this to “the first neural nano-optics imager”.

Comment: There are quite a lot of descriptions which may be wrong, for example, “first”, “widest”. The numerical aperture (the author uses f-number) of the metalens could be very high.

Response: While higher numerical aperture and wider diameter metalenses have separately been shown, these prior demonstrations were all either severely limited in field of view or bandwidth. There is no existing solution in the literature that simultaneously demonstrates a broadband metalens with an aperture and field of view as wide as ours, at such a low f-number and that accommodates the full visible spectrum. We quantify this performance enhancement in our Supplemental Document in Table S2, comparing against previously demonstrated broadband metalenses. To the best of our knowledge, we believe that we are the first to demonstrate high-quality imaging with a metasurface at the combination optics at 500 μm , which is also the widest aperture for this application in the literature.

Comment: The wavelength dependent focal efficiency should be measured in the visible spectral regime.

Response: We agree that this is an important metric and have now added a figure to our Supplemental Document (Section “Validation of Neural Nano-Optic Design”, Figure S15) that compares the diffraction efficiency of our design to that of several baseline designs. The proposed end-to-end design process learned a design with slightly reduced focusing efficiency compared to the baselines, demonstrating that focusing efficiency is not a necessary merit for high-quality imaging. Nevertheless, if one wanted to hand-craft a design with improved focusing efficiency the framework allows for this by incorporating additional constraints in our optimization routine to maximize the enclosed power within the PSF. We have included additional results in Figure S15 that demonstrate how incorporating these constraints increases the focusing efficiency.

References:

1. Engelberg, J. & Levy, U. The advantages of metalenses over diffractive lenses. *Nature Communications* 11 (2020).
2. Mait, J. N., Athale, R. A., van der Gracht, J. & Euliss, G.W. Potential applications of metamaterials to computational imaging. In *Frontiers in Optics / Laser Science, FTu8B.1* (Optical Society of America, 2020).

REVIEWERS' COMMENTS

Reviewer #1 (Remarks to the Author):

The authors have done an excellent job in clarifying certain aspects of the original manuscript. I recommend publication in the current form.

Reviewer #2 (Remarks to the Author):

The revision has addressed most of my concerns and comments reasonably.

Reviewer #3 (Remarks to the Author):

I am happy with the current version and have no additional comments.